# Experimental identification of individual insect visual tracking delays in free flight and their effects on visual swarm patterns

**Md. Saiful Islam**◉*, **Imraan A. Faruque**

School of Mechanical and Aerospace Engineering, Oklahoma State University, Stillwater, Oklahoma, United States of America

\* saiful.islam@okstate.edu

**Data Availability Statement:** All trajectories and stimulus files are available from the figshare database (https://doi.org/10.6084/m9.figshare.

## Abstract

Insects are model systems for swarming robotic agents, yet engineered descriptions do not fully explain the mechanisms by which they provide onboard sensing and feedback to support such motions; in particular, the exact value and population distribution of visuomotor processing delays are not yet quantified, nor the effect of such delays on a visually-interconnected swarm. This study measures untethered insects performing a solo in-flight visual tracking task and applies system identification techniques to build an experimentally-consistent model of the visual tracking behaviors, and then integrates the measured experimental delay and its variation into a visually interconnected swarm model to develop theoretical and simulated solutions and stability limits. The experimental techniques include the development of a moving visual stimulus and real-time multi camera based tracking system called VISIONS (Visual Input System Identification from Outputs of Naturalistic Swarms) providing the capability to recognize and simultaneously track both a visual stimulus (input) and an insect at a frame rate of 60-120 Hz. A frequency domain analysis of honeybee tracking trajectories is conducted via fast Fourier and Chirp Z transforms, identifying a coherent linear region and its model structure. The model output is compared in time and frequency domain simulations. The experimentally measured delays are then related to probability density functions, and both the measured delays and their distribution are incorporated as inter-agent interaction delays in a second order swarming dynamics model. Linear stability and bifurcation analysis on the long range asymptotic behavior is used to identify delay distributions leading to a family of solutions with stable and unstable swarm center of mass (barycenter) locations. Numerical simulations are used to verify these results with both continuous and measured distributions. The results of this experiment quantify a model structure and temporal lag (transport delay) in the closed loop dynamics, and show that this delay varies across 50 individuals from 5-110ms, with an average delay of 22ms and a standard deviation of 40ms. When analyzed within the swarm model, the measured delays support a diversity of solutions and indicate an unstable barycenter.

19493642.v1; https://doi.org/10.6084/m9.figshare.19493597.v1).

**Funding:** Faruque, Office of Naval Research, N0014-19-1-2216, www.onr.gov The funders had no role in study design, data collection and analysis, decision to publish, or preparation of the manuscript.

**Competing interests:** The authors have declared that no competing interests exist.

**Abbreviations:** $A_a(x_{ij}, \tau_{ij})$, Attractive potential; $A_r(x_{ij}, B_r, C_r)$, Repulsive potential; $B_r$, Amplitude of the repulsive potential; $C_r$, Applied distance of repulsive potential; $d$, Number of estimated parameters; $D_1$, Stimulus trajectory; $D_2$, Insect trajectory; $e(s)$, Tracking error; $e(t, \theta)$, Prediction error; $E_t$, Background of the current frame; FIT, Final prediction error; $F_t$, Current frame; $G(s)$, Transfer function; $G_e(s)$, Estimated transfer function; $I$, Magnitude of movement; $J$, Jacobian matrix; $K$, Foreground of the current frame; $M(s, \tau_i)$, Delay model; MSE, Mean square error; $N$, Number of agents; $n_\lambda$, Coefficient of inhibition; $n_\alpha$, Threshold for segmentation; $n_\gamma$, Tracking threshold; $n_\mu$, Background weight; $n_\psi$, Tolerance of error of the re-projected points; $n_a$, Number of poles in the transfer function; $n_b$, Number of zeros in the transfer function; $n_h$, Number of image pixels in horizontal direction; $n_v$, Number of image pixels in vertical direction; $P$, Calibration matrix of camera; $R_C$, Center of mass of the swarm; $S$, Tracking section; $S_x(s)$, Power spectral density of signal $x$; $S_{xy}(s)$, Cross power spectral density of signal $x$ and $y$; $v_i$, Velocity of the swarm agent; $V_m$, Velocity convergent; $w(t)$, White noise; $X$, X axis of the world frame; $x_i$, Position of the swarm agent; $X_i$, Sum of distance from agent $i$ to all other agent $j$; $x_{ij}$, Inter agent distance between agent $i$ to agent $j$; $X_m$, Finite displacement; $Y$, Y axis of the world frame; $y_m$, Model predicted output; $\mathcal{N}$, Gaussian distribution; $\hat{H}(s)$, Measured frequency domain data; $\bar{x}_i$, Filtered position of agent; $\Delta(T)(m)$, Absolute difference between two point pairs; $\gamma^2(s)$, Coherence signal; $\delta t$, Sampling time interval; $\theta$, Estimated transfer function parameters; $\rho$, Coupling strength; $\tau_{ij}$, Time delay from agent $i$ to $j$; $\varphi$, Regression matrix; $\epsilon_a$, Standard deviation of acceleration; $\epsilon_x$, Standard deviation of position in horizontal axis; $\epsilon_y$, Standard deviation of position in vertical axis.

## Introduction

Insects are model systems for resource-constrained micro air vehicles operating in group and swarm applications. Although these naturalistic swarms rely on limited sensory and neural feedback structures and lack a traditional engineered communication network, they achieve coordinated flight maneuvers in near proximity to neighbors and amidst changing neighbors in unstructured environments [1–3]. Vision is one of the few sensing modalities whose quantified bandwidth, range, and sensitivity could provide realtime feedback to modulate these flight paths, and the relatively large fraction of insect neural material dedicated to visual processing suggests that visual control may be an important tool for implicit communication [4–6]. The exact mechanisms that are used to facilitate this interconnection are not yet known. Many theoretical swarm models and analyses remain disconnected from experimental studies on naturalistic swarms, which limits the ability of these parallel efforts to inform each other [7, 8]. In particular, the effect of latencies such as sensory processing delay in biological (or computational in robotic) swarm experiments has not been adequately connected to theoretical developments.

This study develops an experimental tool based on a flight arena equipped with a visual stimulus system (Stimulus design) and a real-time insect tracking routine called Visual Input System Identification from Outputs of Naturalistic Swarms (VISIONS) for insects in free flight (Tracking system), and uses this tool to experimentally identify the visual processing and feedback delay across different insects. The identified delays are measured at the individual insect level, and probability distributions are experimentally quantified for the measured delays ((System identification). The swarm level effect of these delay distributions are then connected to recent progress in theoretical swarm communications analysis and a implemented in a simulated swarm environment (Analysis & simulation). The analysis and simulation indicate that three characteristic patterns may be generated by a collection of insects visually navigating relative to each other with such delays, and identify the Gaussian delay distribution range needed to generate stable swarm (Results and discussion).

## Previous work & background

### Experiments for insects' visuomotor response

The effects of environmental stimuli examining insect flight behavior and motions during visually-dominated behaviors like obstacle avoidance, landing on a wall or proboscis, and flower tracking have previously focused most on the role of ambient and external illumination levels [9–11]. By high-speed videography, it was observed free-flying fruit flies generating quick turns through modest wing alterations and discussed methods for a one-parameter open loop paradigm for free-flying insects under a visual display [12]. To adapt for rapid changes in body orientation, dipterans and bees adjust the amplitude and angle of their wing stroke asymmetrically [13–15]. Nocturnal sweat bees do not change their flight speed while landing in a tunnel [16]. Later work showed that diurnal insects like bumblebees and hornets reduce their flight speed and take progressively more circuitous paths as the light intensity decreases [17], however nocturnal bees *Megalopta* were able to maintain flight speed with decreasing light intensity, including a spatial summation mechanism to explain their nighttime heightened sensitivity [18]. Flower tracking investigations on *Manduca sexta* utilizing a moving flower at various frequencies indicate tracking performance during feeding and energy expenditure, revealing that flower movement direction has a stronger effect on tracking performance than the target's movement frequency [19]. When flying in the wind, the unsteady flight of a hawkmoth in the wake of a 3D printed robotic flower displays larger tracking overshoot and a reduced order dynamic system [20] and a flower tracking experiment was used to quantify the

change in their flight behavior under various light conditions, with flower tracking behavior represented by a simple temporal delay at various light intensities [21]. By adjusting light intensity, a system identification approach was utilized to find a brightness-dependent delay term in the transfer function, resulting in a dynamic model for each Hawkmoth variant that included a combination of species-dependent scaling parameters and processing delays [22]. There is evidence to suggest the internal delay time may have a dependency on swarm sizes in mosquitoes tracking a moving marker [23] and by task; in robber fly predation and obstacle avoidance tasks, different time delays (30 and 90 ms) for proportional navigation and obstacle avoidance rule were observed [24].

While experimental research is beginning to understand the need to quantify internal delays due to insect sensing and feedback, these previous studies are limited to reporting a single average delay across all animals and do not yet account for the heterogeneity of delay across the population or the effect of such delays on neighbor-coordinated behaviors.

## Camera based tracking technologies

Animal movement study has benefited greatly from availability of electronic cameras to reconstruct animal positions. Early automated tools for tracking insects like "Ethovision" and "Trackit" relied on color cameras and chrominance differences to segment foreground and background [25, 26]. "Flydra" achieved real-time tracking and allowed the use of monochrome cameras without the need for chrominance-based segmentation by assuming a fixed background scene and using intensity based segmentation [27]. This technology began to move to other schooling animals, with parallel improvements in outdoor starling tracking applying trifocal cameras began to bring this technology outside the lab, assuming homogeneous sky through a fixed background assumption [28]. Other animals began to be tracked, like fish and mice position tracking from post-processed videos with "IdTracker", [29], and automotive traffic [30], both relying on a static background assumption. "Toxtrac" tracked salmon, zebrafish, and cockroaches using a fixed background assumption, and similarly relied on a static background with a well-lit subject [31], as did infrared tracking approaches on ants at 2 Hz update rate [32]. Further, visual tracking conducted in a naturalistic settings often results in lighting changes that negatively affect tracking approaches [29, 31].

For studies involving an insect or collection of insects tracking a visual stimulus, the appearance of visual stimulus in the background is necessary for precise timing measurements (as will be discussed in the next subsection). Recent works allow limited noisy backgrounds [33, 34], and many tracking problems are still completed manually [35]. Animal flight studies have progressed with the help of high-speed imaging systems that allow researchers to monitor the detailed kinematic behavior of animals during flight [36]. Using artificial markers on the wings wing deformations can be measured and quantified with high speed videography [37, 38]. Because of the small wing sizes, rapid frequency flapping, and unpredictable movements in near proximity to the insect body, wing kinematics characterization and reconstruction remain a challenging task even with high-speed videography [39, 40]. Recent improvements in high speed visual tracking digitizing wing motions have integrated mechanisms to handle dynamic backgrounds and occlusions [41] such as the moving average [42] and Kalman filter [43, 44] techniques, and these approaches have each shown value in real-time visual tracking for robust visual tracking system identification.

## System identification

System identification techniques are applicable to experimental manipulations in which an input and output stimulus pair must be used to discover a set of unknown underlying

dynamics. These experiments generally involve prescribing inputs that perturb a system (sometimes referred to as a 'black box') with known input signals and recording the resulting output variables [45]. This identification problem is often solved in both time and frequency domains [46]. Frequency domain analysis includes the ability to generate non-parametric identifications, applicability to classical control-system design methodologies and modeling of flying qualities, noise robustness, and reduced dimensionality for model parameter estimations [47, 48]. For a linear dynamic model estimate, a diverse set of methods based on frequency domain identification are available, such as equation error [49], filter and output error [46], and spectral estimation [50]. The relative maturity of such models has resulted in applications to insect flight behaviors such as chasing mates [51], saccade turns [52], obstacle avoidance [53], speed control [10], navigation [54], and flight stabilization [55–57].

## Theoretical swarm models with delay

**Swarm models.**  Mathematical descriptions of collective motions of multi-agent biological systems, including bacterial colonies, slime molds, locusts, and fish, have become available in the last few decades [58–60]. These models range from continuous approximations and kinetic theory models to models at the individual level, with interaction mechanisms generally consisting of attractive and repulsive forces [61–63] that move the particles. Aerial insect sensory systems are implemented at the individual level, which lends itself to treating each biological or mechanical individual as a discrete particle. This structure is consistent with individual models like the evolvable heading Vicsek [64] and evolvable-velocity Cucker-Smale [65] models. Both of which are composed of a set of differential equations (each representing a single agent's dynamics) networked together by some idealized sensing and interaction rule. Rigorous proofs are available to show sufficiency of particular rules to lead to flocking-like behaviors showing velocity convergence.

**Bifurcation analysis.**  In differential equation analysis, a bifurcation parameter can be used to understand when a dynamic system exhibits distinct changes in behavior, such as stability or topological structures such as saddle points [66]. Modulating a single bifurcation parameter may achieve distinct behavioral characteristics using pitch fork, Hopf, and transcritical bifurcation types [67]. Different swarming patterns of spacecrafts by attraction and repulsive potentials through dynamic system theory can be obtained [68, 69].

**Recent progress on theoretical delays.**  Despite the availability of rigorous proofs for sufficiency (e.g. [65]), the theoretical models have often relied on a high level of instantaneous connectivity that may be impractical in nature or robotic implementations. The theoretical effects of delay in such a network of interaction rules has previously been investigated primarily numerically by distributing delays among agents and applying a mean field analysis [70]. Only recently are these effects beginning to be understood theoretically by extending the Cucker-Smale model to include the effects of time delay, which provides a path to understanding the effects of sensor and communication delays in swarm networks. Communication weights and a Lyapunov approach are used to determine an upper bound on time delays for velocity convergence for the Cucker Smale model [71].

An upper bound on the size of time delays is now available as a sufficient condition for flocking behaviors [72]. These effects of fixed time delays have recently been applied to both first and second order swarm models with fixed time delays [73] in theoretical and numerical simulation. Bifurcation analysis applied to the swarm network problem with delay as the bifurcation parameter can yield insights into the delay structures that affect collective motion behaviors [74]. To overcome the existing mean field theory's predictability limitations, stability

analysis of ring and rotating patterns in a delay coupled swarm is carried out utilizing the bifurcation parameter [75].

## Contribution of this paper

Despite the progress in different tracking experiments on animals, there is a lack of archival literature directly connecting experimentally measured delays in swarming-capable animals to the theoretical structures available. It remains unknown to what degree visuomotor tracking delays vary across individuals in a population (or even whether they might be better be modeled as fixed or heterogeneous), and whether these individual delays support or conflict with current understanding of visual swarming. A primary contribution of this work is to address these gaps between theory and experiment by providing: (a) an experimental quantification of how individual insect visuomotor delays vary over population, and (b) a theoretical and simulation analysis of the effect of these delays on visually connected swarms, illustrating the achievable shapes and stability regions and the need for delay modeling.

## Materials and methods

### Experimental data collection

We prepared a rectangular acrylic enclosure of $15 \times 10 \times 10$ inch dimensions, as illustrated in (Fig 1a and 1b). Honeybees were collected from a research beehive and placed inside the enclosure at 72°F. The experiment required constructing two pieces of equipment: a visual stimulus input and a real-time tracking system. The experiment moves the stimulus in the $X$ axis of the world coordinate, while the insect flying in the working volume tracks it and we record its trajectories by the tracking system. The insects track the light intermittently rather than continuously (S1 Video), and this visual tracking study analyses the periods of tracking. Tracking sections $S$ in a trajectory were identified automatically as follows,

$$S = [\Delta T(q)...\Delta T(q+g)], \ g \geq 30, \ q = 1,...,k-30 \,, \tag{1}$$

where k is the total number of points. $\Delta T(m)$ is the absolute difference of every pair of points defined by $\Delta T(m) = |T_1(m) - T_2(m)| < \eta_\sigma$. $T_1(m)$ and $T_2(m)$ are continuous trajectory point $m$ of X coordinates of the stimulus and bee. The tracking tolerance $\eta_\gamma$ was used to specify the desired tracking accuracy. $S = [S_1^1, S_2^1, ...S_g^1 \quad S_1^2, S_2^2, ...S_g^2]$ is the identified section containing $g$ tracking point pairs where $S_{(1...g)}^1$ and $S_{(1...g)}^2$ are stimulus and bee points respectively.

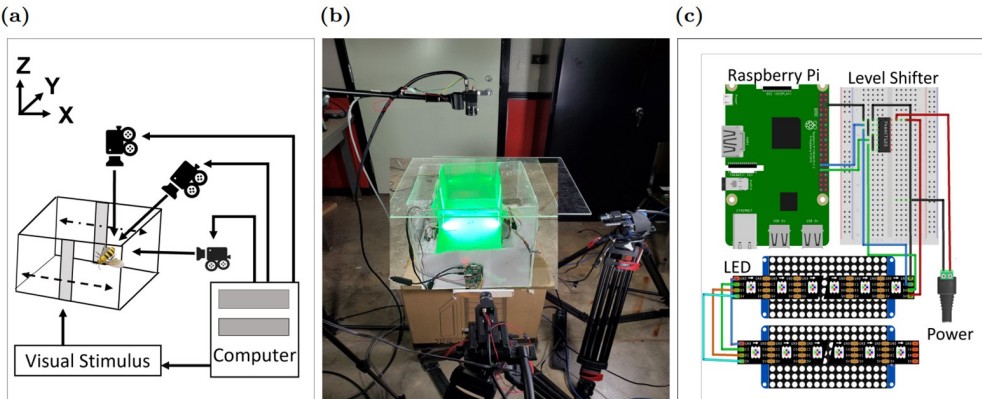

**Fig 1. (a) Diagram of tracking system, (b) physical implementation, (c) input stimulus circuit design.**

## Stimulus design

The visual light stimulus is provided via a two-sided rectangular LED display, each of them contains 16 by 32 (width × height) LEDs with a refresh rate of 100 Hz. A Linux computer and a Raspberry Pi microcontroller are used to generate a programmable image on the LED panel. Fig 1c depicts the two Adafruit dotstar LED displays, one 74aN174 level shifter, and a Raspberry Pi microprocessor, as well as their wiring. Our stimulus was a synchronized vertical green light with a height of 2 by 8 inches (width×height), and a brightness of 2200 lux (measured at the center point between the two opposing lights), as seen in Fig 1b. The stimulus is moved by a sum of sinusoidal frequencies that changed the location of the vertical bar along the world coordinate's X axis.

## Tracking system

Three synchronized cameras (Flea mono cameras) record the bees' flight paths at 50–120 frames per second. Fig 1a shows the cameras set on three tripods with angular separations varying from 50 to 90 degrees. The raw images (1280x1024) collected by these cameras are transported to a central Linux computer through USB3 connections. The ROS platform's Python-based program runs in real time and may also be used for recording and analysis offline. The main flowchart for the tracking system is shown in Fig 2, and the major processes are discussed below.

**2d centroids detection.** After obtaining raw photos from the cameras, we distinguish the foreground from the background. When the background varies slightly, the moving average technique is appropriate. The initial background is found by taking average of first few frames. Then, the background $E_t(i, j)$ for each new frame at time $t$ is updated as

$$E_t(i,j) = (1 - \eta_\mu)E_{t-1}(i,j) + \eta_\mu F_t(i,j), \tag{2}$$

where $i$ and $j$ denote the pixel coordinates that change in every frame, and $F_t(i, j)$ is the current frame at time $t$. The background change is controlled by the weight $\eta_\mu$, which determines how much the background of the next frame should differ from the present one. Calculating the binary difference between the current frame $F_t(i, j)$ and the estimated background $E_t(i, j)$ yields

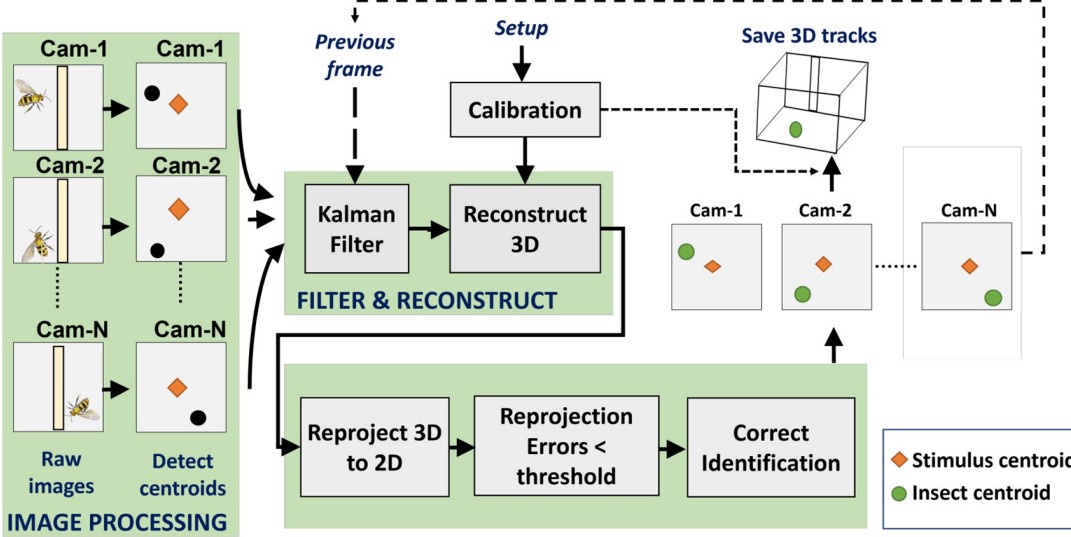

**Fig 2. Flowchart of tracking system.**

the foreground by

$$K(i,j) = \begin{cases} 0, |F_t(i,j) - E_t(i,j)| < \eta_\alpha \\ 1, |F_t(i,j) - E_t(i,j)| \geq \eta_\alpha \end{cases}. \tag{3}$$

If the difference between each pixel coordinate of $F_t(i,j) - E_t(i,j)$ is smaller than the predefined threshold value $\eta_\alpha$, we will consider that pixel point to be zero (dark) in image $K(i,j)$. The threshold $\eta_\gamma$ may be automatically determined by

$$\eta_\alpha = \eta_\lambda \cdot \frac{1}{n_h \cdot n_v} \cdot \sum_{i=1}^{n_h} \sum_{j=1}^{n_v} |F_t(i,j) - E_t(i,j)|, \tag{4}$$

where $n_h$, $n_v$ are the number of image pixels in horizontal and vertical directions of the image and $\eta_\lambda$ is the coefficient of inhibition. The value of $\eta_\alpha$ ranges between 10 and 30. The images may still have some noise. To get rid of the unwanted elements "Gaussian blurring" and "morphological closure" are applied by opencv image processing toolbox where Gaussian kernel acts as a low pass filter to eliminate high frequency components and morphological closure helps to fill small gaps in the images. After the stimulus and insect have been segmented, we calculate the area size, two-dimensional center position (centroid), shapes, and other metrics. Occasional frames may be missed owing to overlaps or occlusion. Using the discrete Kalman filter [43], each insect centroid position can be predicted when there are any missing frames. Detailed information about the Kalman filtering method is described in appendix (section Kinematic filtering). These 2D centroids are required in the next stage.

**2D to 3D conversion.** 3D position from the 2D measured positions of all cameras can be calculated by camera calibration matrices. The intrinsic and extrinsic calibration parameters for each of camera are found using the Svoboda multi camera calibration toolbox [76]. During the Matlab execution of this toolbox, a red laser light in the working volume must be moved. Using these calibration matrices and previously tracked 2D points, we employ the linear triangulation approach to obtain 3D points. For each camera model $j$, the linear triangulation equation is stated as follows

$$\begin{bmatrix} v_i^j P_3^j - P_2^j \\ u_i^j P_3^j - P_1^j \\ .... \\ .... \end{bmatrix} O_i = 0, \tag{5}$$

where $(u_i^j, v_i^j)$ is 2D point $i$ of $j^{th}$ camera, $O_i$ is the 3D point and $P_n^j$ is the $n^{th}$ row of the calibration matrix $P_{3 \times 4}$ of camera $j$. Combining perspective projection expressions for multiple cameras yields a homogeneous system of linear equations; at least two cameras are required to solve this system of equations via singular value decomposition. The combination of 2D points acquired from the cameras can be used to calculate 3D points. The 3D point then is re-projected back to the 2D point by the calibration matrices. A tolerance $\eta_\psi$ on the re-projection error between the true and re-projected 2D points is used to specify the desired precision.

## System identification

In the frequency domain, tracking trajectories may be described by gain, phase, and coherence. The gain depicts the insect's amplitude in relation to the stimulus in each frequency component. To translate the trajectory from time domain to frequency domain, we apply the Fourier

transform (FFT). Tracking error $e(s)$ may be computed by reference to ideal tracking (gain 1, phase 0) as

$$e(s) = \|G(s) - (1 + 0j)\|, \tag{6}$$

where $G(s)$ is the transfer function containing gain and phase for each frequency $s = j\omega$ point. Coherence $\gamma^2(s)$ is calculated from the spectral power density of an input and output signal pair $(x, y)$ by the following equation as

$$\gamma^2(s) = \frac{|S_{xy}(s)|^2}{S_x(s)S_y(s)}, \tag{7}$$

where $S_{xy}(s)$ denotes the cross spectral power density and $S_x(s)$ and $S_y(s)$ represent the auto power spectral density of the stimulus (input) and bee (output) coordinates, respectively. The coherence between the stimulus and the bee trajectory is used to determine the degree of linear connection throughout the frequency range with 1 denoting a purely linear system relationship and 0 denoting no linear relationship. For this investigation, we assumed a linear relationship between the stimulus and the bee trajectories up to a coherent frequency with a coherence value greater than 0.7.

The FFT transform outputs a discrete time frequency domain signal up to the Nyquist frequency ($\frac{1}{2\delta t}$). To improve the resolution over a targeted frequency range identified from the coherence, Chirp Z transform (CZT) [77] was applied. CZT is a generalization of discrete Fourier transform (DFT) and samples the Z plane via spiral arcs (straight lines in the S plane), whereas the DFT samples the complex Z plane at equally spaced points along the unit circle. We consider this CZT frequency domain data of the stimulus $D_1(s)$ and bee $D_2(s)$ trajectories to examine the flight dynamics of the tracking trajectories. We conduct the system identification technique across several possible pole zero combinations (2–4 poles and 1–4 zeros) of transfer functions $G_e(s)$ and varying time delays $\tau_i \in [0, 200]$ ms. A processing delay $\tau_i$ could be modeled as a pure tracking delay ($e^{-\tau_i s}$) or linear approximation $\left(\frac{1}{1+\tau_i s}\right)$ in the transfer function. We included both delay structures in the system identification framework to compare the results. The identified transfer function model is found from the minimum absolute difference between true and model transfer functions over region of coherence, which is presented as

$$\min_{\tau_i, G_e} |\hat{H}(s) - G_e(s)M(s, \tau_i)|_{s=j\omega, \ \omega=\arg\{\gamma^2(s)>0.7\}}, \tag{8}$$

with delay model structure $M(s, \tau_i) = \{e^{-s\tau_i}, \frac{1}{1+\tau_i s}\}$. Here, $\hat{H}(s)$ represents the measured frequency domain data that was derived by dividing the frequency domain versions of the bee and stimulus trajectories, denoted by $\hat{H}(s) = \frac{D_2(s)}{D_1(s)}$. The system identification method is depicted as a flowchart in Fig 3. Three fit criteria FIT, MSE (mean square error) and FPE (final prediction error) [45] are used to determine the best dynamics model. The FIT and MSE criteria are defined by

$$\text{FIT} = 100 \times \left(1 - \frac{\|y - y_m\|}{\|y - \text{mean}(y)\|}\right), \tag{9}$$

$$\text{MSE} = \frac{1}{n}\sum_{i=1}^{n}[y(i) - y_m(i)]^2. \tag{10}$$

where $y$ is the true output, $y_m$ is the model's predicted output and $n$ is number of data points.

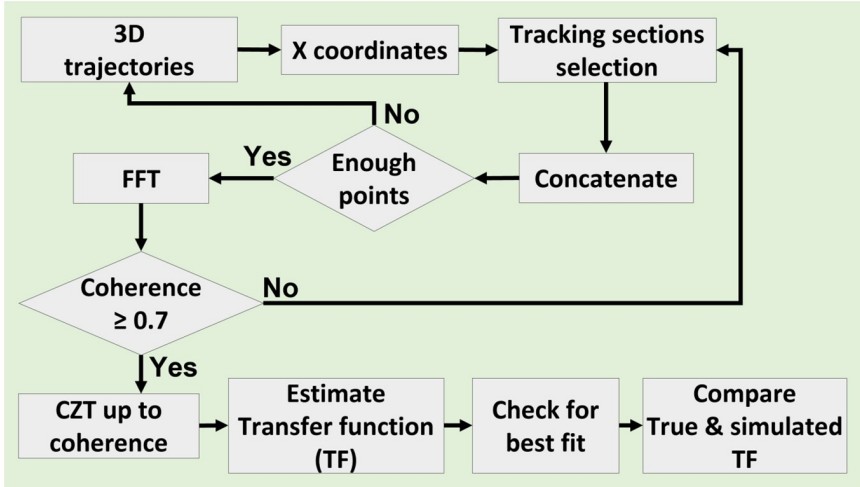

**Fig 3. Flowchart of system identification.**

To find the FPE, we have considered an ARX (auto-regressive with exogenous input) model as

$$y(t) + a_1 y(t-1) + .... + a_n y(t-n_a) = \quad b_1 u(t-1) + ... + b_n u(t-n_b)... + w(t), \quad (11)$$

where $n_a$, $n_b$, $w(t)$ are the number of poles, number of zeros and white noise term respectively. The estimated parameters are $\theta = [a_1, a_2, ...., a_n b_1, b_2, ...., b_n]^T$ and regression matrix is $\phi = [-y(t-1).....-y(t-n_a)u(t-1).....u(t-n_b)]^T$. The prediction error is $e(t, \theta) = y(t) - \phi^T \theta$. FPE is described by

$$\text{FPE} = \det\left(\frac{1}{L}\sum_1^L e(t,\theta)e(t,\theta)^T\right)\left(\frac{1+\frac{d}{L}}{1-\frac{d}{L}}\right), \quad (12)$$

where $L$ is the number of values in the estimate data set and the number of estimated parameters is represented by $d$.

## Swarm model

Our experiment extracts single bee tracking dynamics, and we build a hypothetical swarm of visually interconnected insects from those delays and transfer functions. Each swarm agent individual has a distinct communication latency, consistent with the individual reaction time measurements. A delay-based factor among the insects may impact group behaviors, and we included a coupling strength to allow one to consider multiple swarm shapes.

## Swarm dynamics

We consider a second order (position and velocity) dynamic system model for $N$ agents communicating with each other agents with some heterogeneous processing delays. These delays are heterogeneous i.e, each agent has distinct delay. The dynamic model based on 2D or 3D position $x_i$ and velocity $v_i$ may be written in either as

$$\frac{dx_i}{dt} = v_i, \quad (13)$$

$$m\frac{dv_i}{dt} = -\nabla_i A^a(x_{ij}, \tau_{ij}) - \nabla_i A^r(x_{ij}, B_r, C_r) - \beta v_i, \tag{14}$$

where $\nabla_i A^a(x_{ij}, \tau_{ij})$ and $\nabla_i A^r(x_{ij}, B_r, C_r)$ are an attractive and repulsive potential respectively, specified as

$$A^a(x_{ij}, \tau_{ij}) = -\frac{1}{2}\rho\left(\sum_{i=1,i\neq j}^{N}(x_i(t) - x_j(t - \tau_{ij})) - r\right)^2 + \frac{1}{4}\left(\sum_{i=1,i\neq j}^{N}(x_i(t) - x_j(t - \tau_{ij})) - r\right)^4, \tag{15}$$

$$A^r(x_{ij}, B_r, C_r) = \sum_{i=1,i\neq j}^{N}B_r e^{\frac{-|x_i(t)-x_j(t-\tau_{ij})|}{C_r}}. \tag{16}$$

Here, $\tau_{ij}$ is the delay from agent $i$ to agent $j$. $x_{ij}$ is inter-agent distance of agent i to agent j with some time delay $\tau_{ij}$ and is denoted by $x_{ij} = x_i(t) - x_j(t - \tau_{ij})$. $\rho$ is inter-agent coupling strength and we assume each agent goes to $r$ distance from the origin. $B_r$ and $C_r$ are the amplitude and applied distance of repulsive potential. $\beta$ is the coefficient of friction. For notational convenience, let $X_i = \sum_{i=1,i\neq j}^{N}(x_i(t) - x_j(t - \tau_{ij}))$. The magnitude of the movement $I$ is written as

$$I = -\rho\left(\|\frac{1}{N}X_i\| - r\right) + \left(\|\frac{1}{N}X_i\| - r\right)^3, \tag{17}$$

and the attractive potential gradient is

$$\nabla_i A^a = [I\cos(\theta)\ I\sin(\theta)], \tag{18}$$

where $\cos(\theta) = \frac{X_i(1)}{\|\frac{1}{N}X_i\|}$ and $\sin(\theta) = \frac{X_i(2)}{\|\frac{1}{N}X_i\|}$, and $X_i(1)$ and $X_i(2)$ are the first and second coordinates of $X_i$ respectively. The solution $x(t) = \{x_1(t), x_2(t), ...., x_N(t)\}$ and $v(t) = \{v_1(t), v_2(t), ...., v_N(t)\}$ tend to consensus for $X_m(t)$ and $V_m(t)$ which are defined by

$$X_m(t) = \max_{i,j}\|x_i(t) - x_j(t)\| \text{ and } V_m(t) = \max_{i,j}\|v_i(t) - v_j(t)\|, \tag{19}$$

when finite displacement and velocity convergence happens among agents. Mathematical representation of a finite displacement and velocity convergence can be written as

$$\sup_{t>0}X_m(t) < +\infty \text{ and } \lim_{t\to\infty}V_m(t) = 0. \tag{20}$$

The center of mass of the swarm $R_C$ can be defined as

$$R_C = \frac{1}{N}\sum_{i=1}^{N}x_i. \tag{21}$$

To find the position stability of the swarm, the swarm center norm can be taken as $\|R_C\| = \sqrt{R_{CX}^2 + R_{CY}^2}$, where $R_{CX}, R_{CY}$ are the $X$ and $Y$ coordinates of the swarm center respectively.

For the simulation described in Analysis & simulation, each agent's position output $x_i$ then passes through its identified transfer function $G_{ei}(s)$. If the identified transfer function is

$G_{ei} = \frac{a_n s^2 + a_1 s + .. + a_0}{1 + b_1 s + ... b_n s^n}$, then the filtered position $\bar{x}_i(t)$ will be,

$$\bar{x}_i(t) = a_0 x_i(t) + ... + a_n x_i(t - n) - b_1 \bar{x}_i(t - 1) - .... - b_n \bar{x}_i(t - n). \tag{22}$$

**Linear stability.** Stability at long range is characterized by the attraction potention (the underlying mathematical theory to show this is included in Appendix section "Long range attraction"). Linear analysis characterises the system's local stability properties. The long range dynamics (attraction potential) may be rewritten as

$$\frac{d}{dt} \begin{bmatrix} x_i \\ v_i \end{bmatrix} = \begin{bmatrix} v_i \\ -\nabla_i A^a - \beta v_i \end{bmatrix}, \tag{23}$$

at equilibrium $\dot{x}_i = 0$, $\dot{v}_i = 0$, and $v_0 = 0$, $A^a(x_0) = 0$ at the equilibrium point $(x_0, v_0)$. This occurs when $X_i = r$ if $\rho < 0$ and $X_i = r, r \pm \sqrt{\rho}$ if $\rho > 0$. The system eigenvalues may be obtained via the Jacobian matrix $J$, given by

$$J = \begin{bmatrix} 0 & 1 \\ -\frac{\partial^2}{\partial X_i^2} A^a & -\beta \end{bmatrix} = \begin{bmatrix} 0 & 1 \\ J_{21} & -\beta \end{bmatrix}, \tag{24}$$

where $J_{21}$ can take on three values depending on $\rho$ and equilibrium points $X_i$. The constant $r$ in this case represents the distance from the origin.

$$J_{21} = \begin{cases} \rho, & \text{if } \rho < 0 \cap X_i = r \\ \rho, & \text{if } \rho > 0 \cap X_i = r \\ -2\rho, & \text{if } \rho > 0 \cap X_i = r \pm \sqrt{\rho} \end{cases} . \tag{25}$$

When $J_{21} = \rho$, the eigenvalues will be $\lambda = -\beta/2 \pm \frac{1}{2}\sqrt{\beta^2 + 4\rho}$ and $\rho < 0$ is sufficient to ensure the eigenvalues have negative real part and the equilibrium is stable. When $\rho > 0$, at least one positive real eigenvalue exists and the equilibrium is unstable. For $J_{21} = -2\rho$, the equilibrium is $X_i = r \pm \sqrt{\rho}$ and the eigenvalues are $\lambda = -\beta/2 \pm \frac{1}{2}\sqrt{\beta^2 - 8\rho}$. Stability of this equilibrium requires $\beta > 0$ and $\rho > 0$.

The attraction potential can be used as a pitch fork bifurcation equation with $\rho$ as the bifurcation parameter and forms different stable conditions depending on the sign of $\rho$. This is a supercritical pitchfork bifurcation in which $\rho$ bifurcates into two local equilibria from a single equilibrium. Table 1 shows the linear stability based on the sign of $\rho$. This stability analysis finds that there are three possible swarm patterns that may be generated using the bifurcation parameter. The theoretical analysis does not consider the effects of delays. Our expectation is

**Table 1. Equilibrium position and stability.**

| Coupling $\rho$ | Equilibrium $X_i$ | $\frac{\partial^2 A^a}{\partial X_i^2}$ | Stability |
|---|---|---|---|
| $< 0$ | $r$ | $>0$ | Stable min |
| $> 0$ | $r$ | $<0$ | Unstable max |
| | $r+\rho$ | $>0$ | Stable min |
| | $r-\rho$ | $>0$ | Stable min |

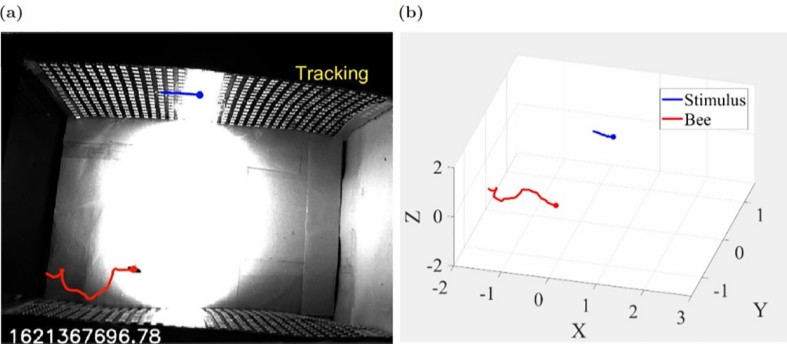

**Fig 4.** (a). Tracking trajectory of bee (red) and stimulus (blue) and "Tracking" text indicates tracking period, (b). their 3d trajectories.

that the time delays as a whole stabilize or destabilize the formation of the swarm shape which is shown in the simulation.

## Results and discussion

An example insect trajectory and its analysis is shown in the experimental section. We have done similar approach to perform system identifications in all other insects. The simulation section demonstrates how changing the bifurcation parameter may produce various swarm configurations. Later, the simulation result demonstrates the impact of identified time delays and transfer functions on the swarm center position.

### Experimental

The input stimulus was generated by eight different sinusoidal signals whose frequency range is 0.1–1.7 Hz and a bee's response is shown in Fig 4. Using the tracking system 3D position data was found. The parameters of the tracking system used in the experiment are shown in Table 2. The bee tracked the light stimulus intermittently. Fig 5a shows stimulus and bee position in 3D and from the X,Y,Z coordinates in Fig 5b it can be seen that the insect tracked the

**Table 2. Properties and thresholds taken for the data collection.**

| $\delta t$ | $\eta_\gamma$ | $\eta_\alpha$ | $\epsilon_x, \epsilon_y, \epsilon_a$ | $\eta_\mu$ | $\eta_\psi$ |
|---|---|---|---|---|---|
| 0.02 | 0.1 | 10 | 0.5 | 0.1 | 0.5 |

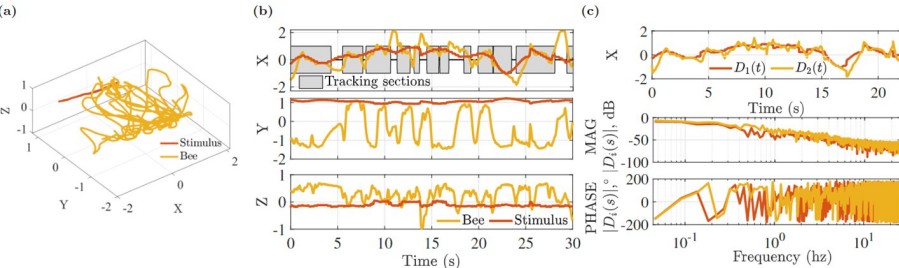

**Fig 5.** (a) 3D plot of bee and stimulus, (b) X, Y, Z coordinates and tracking sections (gray color), (c) concatenated trajectories, magnitude and phase of $D_i$, $i = 1, 2$.

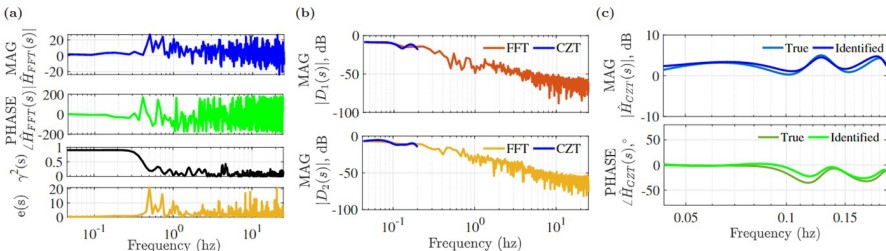

**Fig 6.** (a) Magnitude $|\hat{H}_{FFT}(s)|$ in dB, phase $< \hat{H}_{FFT}(s)$ in °, coherence $\gamma^2(s)$ and tracking error $e(s)$, (b) FFT and CZT magnitude plots of $D_1(s)$ and $D_2(s)$, (c) magnitude and phase of $\hat{H}_{CZT}(s)$.

X axis stimulus movement. Tracking sections are shaded in gray. After obtaining the tracking sections of X coordinates we concatenated them and performed frequency domain transformation. For this trajectory, the concatenated trajectories of stimulus $D_1(t)$ and bee $D_2(t)$, and their magnitude $|D_i(s)|$, dB and phase $\angle D_i(s)$ converted by the FFT transform are shown in Fig 5c. Now we want to see the transfer function of the system by considering stimulus $D_1(s)$ as input and $D_2(s)$ as output. Its transfer function $\hat{H}_{FFT}(s)$ stays close to zero over a frequency range of 0–0.1Hz. The transfer function will have unity gain if the insect responds to the stimulus in an ideal situation. Gain exceeding 0dB indicates the insect's trajectory overshoots the stimulus movement, while gain of 0dB indicates that the bee travels proportionate to the stimuli. The phase component describes the synchronized activity of the insect and the stimuli. The bee's delay in responding to the target is indicated by a negative phase (lag). Using coherence we can identify tracking performance over a frequency range. The coherence plot of Fig 6a shows that the input and output have a strong linear relationship up to 0.1 Hz (as quantified by $\gamma^2 > 0.7$) and its error $e(s)$ is small ($<$5dB) in this range. The number of points in this range, however, is less than 20. To improve the resolution in the region of high coherence, the CZT transform was used. The CZT frequency upper limit, shown in Fig 6b, was set to 0.2 hz to increase resolution in the dominant frequency range. The frequency response function $\hat{H}_{CZT}(s)$ from the CZT was then used to find the best matched transfer function model.

As seen by the FIT error statistics presented in Table 3, a 2 pole, 1 zero transfer function with 21 ms processing latency (transport delay) was the best fit for this example trajectory. The identified model system transfer function and true transfer function plots for this example are shown in Fig 6c, and measured $y(s)$ and simulated $y_m(s)$ frequency domain output are represented in Fig 7a.

$$G_e(s) = e^{-.021s}\frac{-.285s + 9.204}{s^2 + 2.098s + 8.098} \qquad (26)$$

When a transfer function model was fit to the frequency range of the experimental frequency responses having high coherence, best-fit model in terms of pure tracking delay and linear approximation was seen in Table 4. When identifying the system dynamics, the uncertainty of

**Table 3. Model structure and performance for an example insect (best fit highlighted).**

| Model Order | Fit | FPE | MSE | Delay (ms) |
|---|---|---|---|---|
| 2 poles, 1 zero | 92.95% | $0.1073e^{-3}$ | $9.343e^{-3}$ | 21 |
| 3 poles, 2 zeros | 60.96% | 0.003156 | 0.002878 | 15 |
| 4 poles, 3 zeros | 83.25% | $0.6179e^{-3}$ | $0.5297e^{-2}$ | 2 |

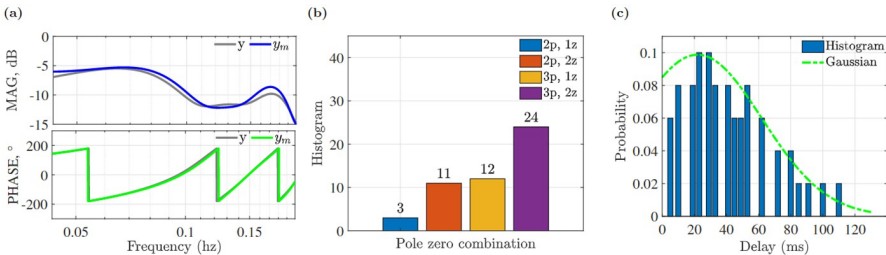

**Fig 7. (a) Magnitude and phase of true $y(s)$ and simulated output $y_m(s)$, (b) histogram of pole zero combination, (c) delay distribution of 50 insects and normalized Gaussian distribution.**

**Table 4. Comparison of pure delay and linear approximation.**

| Delay model | Best FIT trials | Mean FIT | Standard deviation |
|---|---|---|---|
| Pure delay | 36 | 85.45 | 8.54 |
| Linear approximation | 14 | 82.16 | 9.28 |

the transfer function parameters was also obtained. The pure delay model outperformed the linear delay model and was used in the subsequent analysis. The pole zero histogram of the identified transfer functions shown in Fig 7b indicate the three-pole structure was the dominant structure found (82% of insects); and a majority of insects (57%) showed a transfer function with three poles and two zeros. As with the example insect, the identification was insensitive to the choice of fit criteria (eg, max($FIT$), min($MSE$), min($FPE$)) across the 50 insects measured in this study. The individual delay values varied over the dataset, and the relative frequency of the identified delays is shown in Fig 7c.

## Analysis & simulation

When coupling strength $\rho$ is used as a pitch fork bifurcation parameter, a visually-interacting swarm is able to produce both stable and unstable modes, as quantified by linear stability analysis. The achievable parameter values and pattern shapes are summarized in Table 5, which shows the effect of varying $\rho$ on resulting formations (the variation in number of agents $N$ was for visualization and did not affect on the structure).

The visually-interconnected swarm simulations showed a diversity of achievable patterns consistent with theoretical predictions in which $\rho$ was varied to establish the cluster form. The simulations may be conducted in 2D and 3D, with no theoretical impact. Here, we present 2D simulations for computational and presentation simplicity. Parallel 3D cluster formation is illustrated in the appendix (e.g., Fig 8). In both cases, the existence of stable behavior depends on coupling strength $\rho$. The choice of $\rho = -4$ creates a single ring structure. As $\rho$ decreases to 1.5, the ring's stable equilibrium becomes unstable and it bifurcates into two rings. Simulations conducted with the same initial positions Fig 9(a) for 10 seconds and varying parameter values

**Table 5. Multi-agent behaviors formation.**

| Pattern | N | $\rho$ | r | $B_r$ | $C_r$ | $\beta$ |
|---|---|---|---|---|---|---|
| Ring | 30 | -4 | 3 | 1 | 0.5 | 10 |
| Double Ring | 50 | 1.5 | 3 | 1 | 0.5 | 10 |
| Cluster | 50 | -4 | 0 | 1 | 0.5 | 10 |

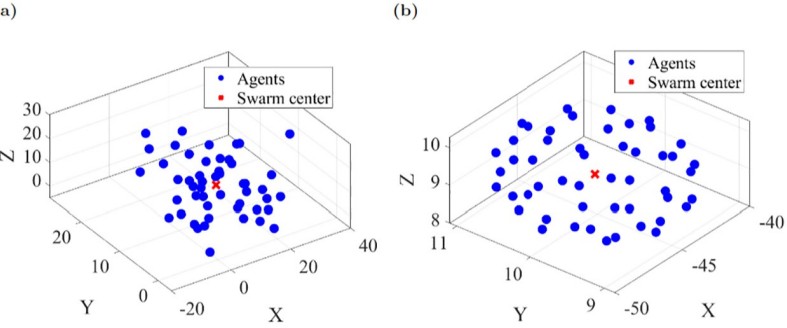

**Fig 8. (a)** Initial condition of 50 agents **(b)** cluster shape formation after 10 seconds.

illustrate this effect, as shown for three cases in S2 Video and (Fig 9b–9d). According to the Eq (20), the simulated agents' position and velocity meet the consensus conditions after a period of time, as seen in (Fig 9e and 9f).

## Effect of identified delays

We investigated the effect of delays on the swarm center. The more naturalistic cluster shape was used to evaluate the swarm's center of mass (barycenter) position stability under varying interaction delays. The simulation was run under different positive interaction delays based on a Gaussian distribution $\mathcal{N}(\tau > 0|\mu, \sigma)$, which is defined as $\mathcal{N}(\tau > 0|\mu, \sigma) = \frac{1}{\sigma\sqrt{2\pi}}e^{-\frac{1}{2}\left(\frac{\tau-\mu}{\sigma}\right)^2}$, where mean $\mu$ and standard deviation $\sigma$ were varied. Plotting the mean $\mu$ and standard deviation $\sigma$ illustrates two distinct stable and unstable areas. As illustrated in Fig 10a, the black points represent the unstable behavior and the green points are for the stable behavior. A third order polynomial was sufficient to describe the boundary between the stable and unstable regions.

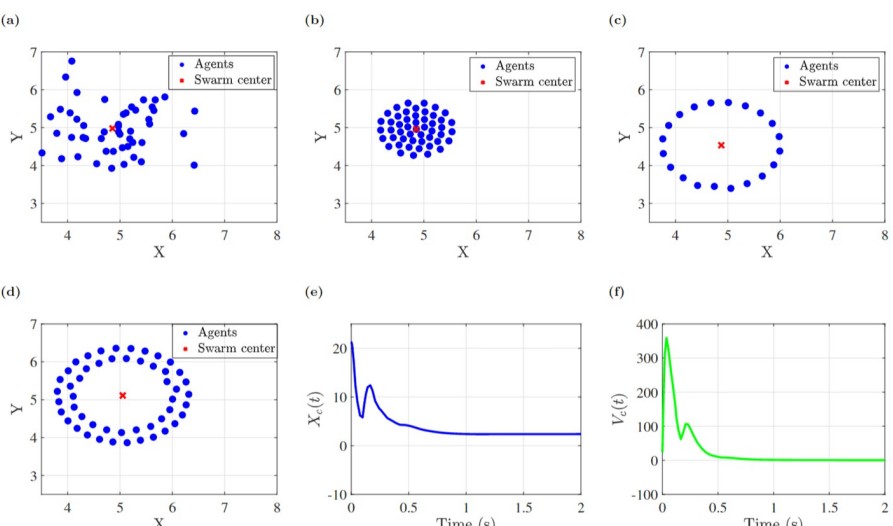

**Fig 9. (a)** Initial position of agents, **(b)** cluster shape, **(c)** ring shape, **(d)** double ring shape, **(e)** and **(f)** consensus of position and velocity for cluster.

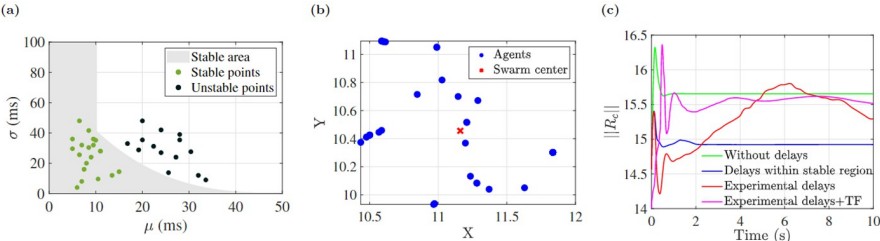

**Fig 10. (a) Gaussian stability area for different mean and standard deviation (gray color indicates stable area, green and black points represent stable and unstable regions respectively), (b) experimental delays show unstable behavior, (c) norm of swarm center $||R_c||$ for different cases.**

In this simulation, interaction delays for each participant in a visually interconnected swarm is assigned a delay randomly from the measured distribution. Because the measured delay histogram approximates a normalized Gaussian distribution with mean $\mu = 22$ and standard deviation $\sigma = 40$ ms located in the unstable region of Fig 10a, our analysis indicates the simulation using measured delays will have an unstable center of mass. We also wanted to simulate using the actual experimental delays. Fig 10b shows the simulation result using the experimental measured delays for 10 seconds. In Fig 10c, the norm of swarm centers $||R_c||$ are shown for all three cases: no delay, Gaussian delays in stable region, and experimental delays. The $||R_c||$ for measured delays shows position instability, while the delay free case and delays within stable area of Fig 10a show position stability of $||R_c||$.

### Effect of both identified delays and transfer functions

We also conducted a simulation including both the identified individual transfer functions as well as the delays, which shows cluster shape formation in the stable region shown in Fig 11. The inclusion of individual agent transfer functions changes the location of the swarm center rather than the overall shape of the swarm. Finally we applied the identified transfer functions and delay variation and its swarm center position is shown in Fig 10c.

### Conclusion

In this study, individual honeybees tracked a visual light stimulus and the visuomotor delay in their closed loop tracking was measured. We developed a real-time camera-based tracking system called VISIONS that track honeybee 3D position and induced them to follow a moving light target. A system identification technique was applied to identify the closed loop tracking dynamics between light stimulus motion and insect body motion, and quantify the delay between the stimulus and animal trajectories, separating the effects of open loop plant

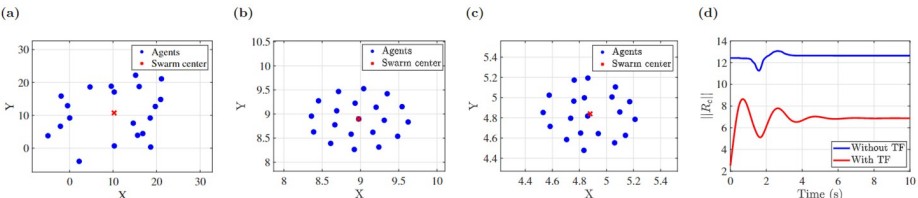

**Fig 11. (a) Initial condition of 20 agents, (b) cluster shape with measured delays, (c) cluster shape with both identified transfer functions and delays, (d) swarm center position.**

(locomotion) from visuomotor feedback dynamics. The measured honeybee sensorimotor delays were used to find a delay distribution across population, showing that insect visual sensorimotor delays in a tracking task are heterogeneous across population.

To understand the implications of the measured delays on visual communication and identified dynamic systems, we then integrated the measured delays and dynamic systems into a visually interacting swarm model. Analysis on this model indicates the range of achievable swarm patterns (cluster, ring, double-ring) and conditions needed for center of mass's position stability of each mode, and simulation illustrates these achievable behaviors and stability regions for both theoretical and measured delays. The analysis and simulation indicate that while the processing delays measured in solitary conditions support three relative shapes, these delays lie in a region associated with an unstable center of mass position and are thus sufficient to support coordinated relative motion but not center of mass position stabilization. This finding suggests that visually interconnected peer insects may be able to to achieve relative configurations but that the measured visual interconnection structure does not support stabilizing the swarm's overall center of mass position. An important distinction is that this study considers conspecific peers and not the effect of other visual targets or stimulus, which may play a role in the group's position. In the absence of external targets or a form of delay modulation (compensation), this analysis and lack of barycenter position stiffness suggests that swarm center of mass position may drift, an important result to completing swarm theory descriptions and informing experiments that investigate solitary and swarm motions in flying insects.

# Appendix

This appendix section includes details of kinematic filtering, long range attraction, and an example of 3D simulation.

## Kinematic filtering

Position $p_k$ and velocity $v_k$ at time $k$ of each agent are

$$p_k = p_{k-1} + v_{k-1}\delta t + \frac{1}{2}a_{k-1}(\delta t)^2, k = 1, 2, 3... \tag{27}$$

$$v_k = v_{k-1} + a_{k-1}(\delta t), \tag{28}$$

where $\delta t$ and $a_k$ are the time step and acceleration respectively. A kinematic model is described as

$$x_k = \begin{bmatrix} p_k \\ v_k \end{bmatrix} = \begin{bmatrix} 1 & \delta t \\ 0 & 1 \end{bmatrix} \begin{bmatrix} p_{k-1} \\ v_{k-1} \end{bmatrix} + \begin{bmatrix} \frac{(\delta t)^2}{2} \\ \delta t \end{bmatrix} a_k. \tag{29}$$

We applied this filter to image coordinates expressed in 2D image frame coordinates. Let $p^x$ and $p^y$ represent the positions, $v^x$ and $v^y$ represent the velocity and $a^x$ and $a^y$ indicate the acceleration in the horizontal and vertical directions of the image frames. Overall, the model may

be expressed as

$$
x_k = \begin{bmatrix} p_k^x \\ p_k^y \\ v_k^x \\ v_k^y \end{bmatrix} = A \begin{bmatrix} p_{k-1}^x \\ p_{k-1}^y \\ v_{k-1}^x \\ v_{k-1}^y \end{bmatrix} + B \begin{bmatrix} a_{k-1}^x \\ a_{k-1}^y \end{bmatrix} ; A = \begin{bmatrix} 1 & 0 & \delta t & 0 \\ 0 & 1 & 0 & \delta t \\ 0 & 0 & 1 & 0 \\ 0 & 0 & 0 & 1 \end{bmatrix}, B = \begin{bmatrix} \dfrac{(\delta t)^2}{2} & 0 \\ 0 & \dfrac{(\delta t)^2}{2} \\ \delta t & 0 \\ 0 & \delta t \end{bmatrix} \tag{30}
$$

$$
x_k = A x_{k-1} + B u_{k-1} ; x_{k-1} = \begin{bmatrix} p_{k-1}^x \\ p_{k-1}^y \\ v_{k-1}^x \\ v_{k-1}^y \end{bmatrix}, u_{k-1} = \begin{bmatrix} a_{k-1}^x \\ a_{k-1}^y \end{bmatrix} \tag{31}
$$

$$
y(t) = C x(t) ; C = \begin{bmatrix} 1 & 0 & 0 & 0 \\ 0 & 1 & 0 & 0 \end{bmatrix}, \tag{32}
$$

where $A$, $B$ and $C$ are state, input and output matrices respectively. Co-variance matrices of process noise $E_x$ and measurement noise $E_z$ are considered statistically by Gaussian noise with normal probability distribution. The process noise covariance matrix $E_x$ can be found from the standard deviation of position and velocity. The standard deviation of position in X and Y direction ($\epsilon_x$, $\epsilon_y$) can be obtained by the standard deviation of acceleration $\epsilon_a$ multiplied by $\frac{(\delta t)^2}{2}$ and the standard deviation of velocity ($\epsilon_{vx}$, $\epsilon_{vy}$) can be obtained by standard deviation of acceleration $\epsilon_a$ multiplied by $\delta t$. $E_x$ and $E_z$ are illustrated as

$$
E_x = \begin{bmatrix} \epsilon_x^2 & 0 & \epsilon_x \epsilon_{vx} & 0 \\ 0 & \epsilon_y^2 & 0 & \epsilon_y \epsilon_{vy} \\ \epsilon_{vx} \epsilon_x & 0 & \epsilon_{vx}^2 & 0 \\ 0 & \epsilon_{vy} \epsilon_y & 0 & \epsilon_{vy}^2 \end{bmatrix} = \begin{bmatrix} \dfrac{(\delta t)^4}{4} & 0 & \dfrac{(\delta t)^3}{2} & 0 \\ 0 & \dfrac{(\delta t)^4}{4} & 0 & \dfrac{(\delta t)^3}{2} \\ \dfrac{(\delta t)^3}{2} & 0 & (\delta t)^2 & 0 \\ 0 & \dfrac{(\delta t)^3}{2} & 0 & (\delta t)^2 \end{bmatrix} \epsilon_a^2, \tag{33}
$$

$$
E_z = \begin{bmatrix} \epsilon_x^2 & 0 \\ 0 & \epsilon_y^2 \end{bmatrix}. \tag{34}
$$

The prediction matrix $P^-(t)$ and integrated $\hat{x}^-(t)$ are applied as

$$
P_k^- = A P_{k-1}^+ A^T + E_x, \tag{35}
$$

$$
\hat{x}_k^- = A \hat{x}_{k-1}^+ + B u_{k-1}. \tag{36}
$$

In the correction step, the Kalman gain $K$ is obtained as

$$K_k = P_k^- C^T (C P_k^- C^T + E_z)^{-1}. \tag{37}$$

Finally, we update the $P^+(t)$ and $\hat{x}^+(t)$ by

$$P_k^+ = (I - K_k C) P_k^-, \tag{38}$$

$$\hat{x}_k^+ = \hat{x}_k^- + K_k(y_k - C\hat{x}_k^-). \tag{39}$$

## Long range attraction

At long distance the repulsive potential have negligible effect on the swarm model. The distance between every agent $i$ to other agent $j$ is taken by $U = |x_i(t) - x_j(t - \tau_{ij})|$. From Eq (14) we can write

$$m \frac{dv_i}{dt} = -\frac{dA^a}{dU} - \frac{dA^r}{dU} - \beta v_i, \tag{40}$$

$$mV \frac{dv_i}{dU} = -\frac{dA^a}{dU} - \frac{B_r}{C_r} e^{-\frac{U}{C_r}} - \beta v_i. \tag{41}$$

Let $S = \frac{U}{r}$, Eq (41) becomes

$$\frac{1}{r} mV \frac{dv_i}{dS} = -\frac{dA^a}{r.dS} + \frac{B_r}{C_r} e^{-\frac{r}{C_r}S} - \beta v_i. \tag{42}$$

At long distance $r >> C_r$, hence $\frac{D_r}{r} \to 0, \frac{B_r}{C_r} e^{-\frac{r}{C_r}S}$ is

$$\lim_{\frac{C_r}{r} \to 0} \frac{B_r}{C_r} e^{-\frac{S}{\frac{C_r}{r}}} = 0. \tag{43}$$

Thus at long distance the repulsive force vanishes, and the attraction potential characterizes long range stability analysis.

**3D simulation.** 3D Cartesian coordinates in Eq 18 must be considered in order to simulate the swarm model in three dimensions. Here, a 3D simulation of 50 agents is illustrated using the parameters $\rho = -4, r = 0, B_r = 1, C_r = .5, \beta = 10$, and random initial conditions. This cluster swarm formation shows that the swarm model works in both 2D and 3D environments.

## Supporting information

**S1 Video. Video of tracking example.** Intermittent tracking trajectory labeled in yellow text.
(MP4)

**S2 Video. Video of pattern shape formation.** Cluster shape, ring, double ring by changing bifurcation parameter.
(MP4)

## Author Contributions

**Conceptualization:** Md. Saiful Islam, Imraan A. Faruque.

**Data curation:** Md. Saiful Islam.

**Formal analysis:** Md. Saiful Islam, Imraan A. Faruque.

**Funding acquisition:** Imraan A. Faruque.

**Investigation:** Imraan A. Faruque.

**Methodology:** Md. Saiful Islam, Imraan A. Faruque.

**Project administration:** Md. Saiful Islam, Imraan A. Faruque.

**Resources:** Imraan A. Faruque.

**Software:** Md. Saiful Islam, Imraan A. Faruque.

**Supervision:** Imraan A. Faruque.

**Validation:** Md. Saiful Islam.

**Visualization:** Md. Saiful Islam.

**Writing – original draft:** Md. Saiful Islam, Imraan A. Faruque.

**Writing – review & editing:** Md. Saiful Islam, Imraan A. Faruque.

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
