## [Decision Letter · Decision Letter 0]

1 Aug 2022

PONE-D-22-10206Experimental identification of individual insect visual tracking delays in free flight and their effects on visual swarm patternsPLOS ONE

Dear Dr. Islam,

Thank you for submitting your manuscript to PLOS ONE. After careful consideration, we feel that it has merit but does not fully meet PLOS ONE’s publication criteria as it currently stands. Therefore, we invite you to submit a revised version of the manuscript that addresses the points raised during the review process. Both reviewers recommended major revision of the manuscript, and gave detailed comments on the points to address in the revision. Please address all of their comments in your revision of this work.

We look forward to receiving your revised manuscript.

Kind regards,

Antony R Humphries, Ph.D.

Academic Editor

PLOS ONE

Journal Requirements:

This work was supported in part by ONR Young Investigator Award N00014-19-1-2216.

However, funding information should not appear in the Acknowledgments section or other areas of your manuscript. We will only publish funding information present in the Funding Statement section of the online submission form. 

Faruque, Office of Naval Research, N0014-19-1-2216, www.onr.gov

Reviewers' comments:

Reviewer's Responses to Questions

**Comments to the Author**

1. Is the manuscript technically sound, and do the data support the conclusions?

Reviewer #1: Partly

Reviewer #2: Partly

2. Has the statistical analysis been performed appropriately and rigorously? 

Reviewer #1: Yes

Reviewer #2: Yes

3. Have the authors made all data underlying the findings in their manuscript fully available?

Reviewer #1: Yes

Reviewer #2: Yes

4. Is the manuscript presented in an intelligible fashion and written in standard English?

Reviewer #1: Yes

Reviewer #2: Yes

5. Review Comments to the Author

Reviewer #1: The topic of the manuscript is of scientific interest, but the

manuscript requires a major revision in my view.

The authors perform systematic experiments with the aim of identifying

visual/motor control delays of bees when bees track a moving visual

stimulus. The delay is part of a complete linear system identification

of the bee (modelled as a transfer function of a linear dynamical

system). The goal is to use the thus identified delay to determine its

influence on swarming and maneuvers of swarms. So, the authors also

insert these identified delays into a bee swarming model and observe

that the resulting delays would destabilize the swarm's configuration.

The current version has a few major problems that should be addressed

in a revision.

Major issues

* It appears that the transfer function is identified in the frequency

domain entirely from frequency components <0.2Hz (so, phenomena on a

time scale of ~5sec or longer). The identified delay is, however,

~40ms (max 100ms). This strikes me as a discrepancy. The errors

measured (3 measures were used, called Fit, FPE, MSE) should be

mapped back into uncertainties of the coefficients in the

identified/fitted transfer function G to demonstrate that the delays

are significantly different from zero (expressed via some

confidence intervals). As the delays are so extremely small, they

would likely be well approximated by a inverse linear expansion of

the exponential function (exp(-s)=1/(1+s)), and could thus be

incorporated into the denominator.

* The connection between the experiments performed to gather data and

the simulations is tenuous. The identified transfer function

for each bee implies a certain (linear) dynamical system (2nd or,

mostly, 3rd order, with delay). But it appears that this linear

model was not used in the model at all except for the delay. It

would also be a stretch to argue whether the delays and transfer

function coefficients measured for an isolated bee (is that

correct?) tracking a light strip can be applied to a bee's dynamics

in a swarm, where it would have to track the various targets

represented by the repulsive and attractive potentials (eqs 24, 25).

A question related to the tenuous connection: why are 2d models used

when the experiment goes to great length to extract a 3d trajectory?

* The description of the data processing steps is exceedingly

difficult to follow. There is a large number of inconsistencies in

notation and there is an unevenness in the level of detail given

(some steps are described in great detail while other, equally

important ones, are jumped over). In my list of minor issues I list

a few of the points where I had difficulty following. The list is

not complete, though, such that the authors should revise this

section aiming for more clarity.

Minor issues:

* p5,eq1: is T_j(m) in R^3 or is this a number? Further, Delta T

appears to be a function of a single integer below eq1, but of g+1

integers in eq1. THis makes the definition of S_{g x 2} unclear.

* In eq2 D(i,j) appears on both sides of the equations, so the

equation should be simplified to D(i,j)=E(i,j). I suspect, however,

that the D(i,j) on the left-hand side is the D(i,j) of the next time

step. It would be helpful to include time as an argument because

that would also clarify if in eq3 the updated D(i,j) is used or the

D(i,j) from the previous time step.

After the explicit formulas (2-4) a single sentence "apply Gaussian

blur and morphological closure" skips over much more substantial

transformations.

* Eq7 has several strange features. It appears to be a differential

equation, dot x(t)=Ax(t-1)+Bu(t), with an enormous delay equal to 1

(measured in seconds?). The quantities p_t and v_t are not explained

and the relationship between x and p,v is not defined. I suspect

that eq7 is not a differential equation but a time step of length

delta t. Correspondingly, should the first term be A x(t-delta t)? A

similar correction may have to be applied to eqs (5,6)?

* The term u in eq7 is never defined. I suspect that the Kalman filter

is applied to the model of straight flight where measurement data

corrects the model (in effect providing the acceleration a(t))? I

could not understand the description. If eq7 is not a differential

equation, one may have to use the Kalman filter for discrete time

stepping.

* In eq15 the notation S_3^j is unclear, S_j appears to be a row of a

3x4 matrix, so is a 1x4 (row) vector. How can one take a power of

this vector? The term P_2^j is not defined at all.

* Fig5c seems to suggest that the windows in which the bee is

determined to be tracking are glued together before further data

processing. Is this justified? This could have a large effect on the

delay estimate.

* The equations on p8 contain several undefined terms: G(s) is the

transfer function of what? Later G(s)e^(st au) is mentioned without

specifying what tau is.

The minimisation in eq18 is not clear: what class of G_e is

minimised over and what is tau?

* which norm is used in eq(19)?

* there is no index i in the terms of the sum in eq20.

* Is the quantity e a function of two variables (t,theta) as suggested

in eq(21) or one variable (t) as seen below eq(21)?

* Below eq(27) X_i is is a vector, as established in the sentence. But

then X_i shows up in the denominator defining cos theta. How do we

divide by a vector here?

* In eq(26) a quantity epsilon shows up, while in eq(24) we have rho. Are they the same?

* In eq(29) sup_{t>0} V_m(t)=0 is the same as demanding that V_m(t)=0

for all t>0 (since V_m is non-negative). Do you mean

lim sup_{t\\to\\infty} V_m=0?

* In eqs(31,32) the undefined quantities V and X show up. Are all

these quantities (U,V,X) implicitly having the index i,j?

* It is unclear how the delays are taken care of in the stability

computation in eq(36). They appear to have taken into account in the

final results (as they make a difference in stability). Again, as

the delays are small, the approximation

x(t-tau) ~ x(t) - tau x'(t)

would likely be accurate compared to the experimental uncertainty.

* For the simulation the delays are assumed to have Gaussian

distribution. This does not make sense as it would imply that delays

could be negative. Sigma not much smaller than mu.

Reviewer #2: The authors measure and analyze the effects of time delays in swarms of insects.

First, they extract a characteristic distribution of delays from individual honeybee trajectories that respond to an applied visual stimulus. Then, they input the measured distribution into a swarming model which shows a variety of collective behaviors depending on coupling and distribution parameters, etc. Overall, the work addresses an important problem (the need for quantifying delays in real autonomous mobile systems and the inclusion of such effects in theoretical swarming models. The experimental portion, in particular, is very interesting and convincing. But, the manuscript needs to be significantly rewritten. Very many sections are unclear and easy to misinterpret, especially in terms of analysis. Specific comments and questions are stated below, which should be addressed before publication:

1. In the section “Kinematic tracking”, the control input u(t) is not defined. It seems to be the acceleration, but this is not explained.

2. Equation (7) is a discrete-time system with time-step $\\delta t$, yes? If so, the time-derivative over-dot is probably a typo? Also, only two spatial dimensions are assumed, why not three? The latter fact is explained later, but commentary should be added in this section to orient the reader.

3. Since you describe noise, Equations (6) and (8) should presumably have additive Gaussian white noise terms for example, per the canonical formulation of Kalman filtering.

4. Where does the state-space noise come from in Equation (9)? There needs to be a noise amplitude somewhere, like in E_{z}.

5. The notation in Equation (11) is confusing. Shouldn't the aprioir estimate at the current time depend on x^{+}(t-\\delta t): the aposteriori estimate at the past time...? Overall the mix of discrete and continuous in this section is unclear.

6. There is a parentheses missing in Equation (13).

7. In Equation (16), the notation is confusing. Is "i" imaginary or "j"?

8. Just above Equation (17), is the assumption of linearity (between visual stimulus and trajectory-response) a good approximation? If so, how was that quantified?

9. Equation (21): This is a general formula. It would be more interpretable, if you used the actual variables in the current fitting/estimation problem.

10. There is no self-propulsion in Equations (22-23). Hence, if no neighbors are present, then the agents *stop* due to friction. Is that right? Without some tendency to keep moving, the “swarming” system seems like an inert, equilibrium system, not an active matter system relevant for swarming…Please comment.

11. What is x_{ij}....? It is defined implicitly, though ambiguously, in Equation (25).

12. In Equation (24), shouldn't each quantity within the "sums" be squared, individually? Ditto with the fourth power. Typically, potential functions (at least in physically-inspired models) are pairwise. This phrasing seems to generate cross terms, which is a bit strange.

13. In the section "Long range attraction,” the derivation seems like an unnecessary use of formalism. The repulsive potential is exponentially damped, whereas attraction (for example) is infinite range-- the polynomial functions grow to infinity as the distance does. I would just state the fact that you are looking in the regime of long-range attraction where pairwise distances are >> Dr.

14. X_{i} being constant means that agent "i" is a constant distance from the swarm center of mass, yes? If so, add some commentary and explanation, etc. near Equation (36).

15. The Jacobain for the full swarm should have N*times the dimension of Equation (36). It seems like you are studying the linear stability with respect to perturbations in each agent (treated separately, as localized perturbations)...Probably, this is the same as a mean-field analysis, which is sometimes accurate and sometimes not. To judge, we need to see comparisons to full-swarm simulations. On that point, see for instance https://link.aps.org/doi/10.1103/PhysRevE.101.042202, which formally analyzes delay effects in self-propelled swarming systems. The authors should consider citing this work since it addresses delay-induced instability in swarms beyond the mean-field analysis that they present, and would give the present work some cover in terms of pointing to more detailed treatment, etc.

16. Where does time delay enter into the picture (in the Linear Stability section)? It is not clear at this point. In fact, time delay will not just change stability of the solutions mentioned, but may change the solutions all together. If this is neglected, fine, but the assumption needs to be stated.

17. The section “Results and Discussion” has almost no commentary. It is disorienting to read. The figures are just plopped down, one after another. Please consider rewriting this section.

18. It seems like time-delay effects were only simulated in the swarming model, not formally analyzed (e.g., in the linear stability calculation). Is that correct? For one, if delay was included in the stability analysis, we should see an infinite dimensional stability spectrum, satisfying a transcendental equation with an infinite number of solutions.

6. PLOS authors have the option to publish the peer review history of their article (what does this mean?). If published, this will include your full peer review and any attached files.

Reviewer #1: No

Reviewer #2: No

---

## [Author Response · Author response to Decision Letter 0]

23 Sep 2022

Response to reviewers

Reviewer 1 

Major issues : 

R1C1: It appears that the transfer function is identified in the frequency domain entirely from frequency components <0.2Hz (so, phenomena on a time scale of ~5sec or longer). The identified delay is, however, ~40ms (max 100ms). This strikes me as a discrepancy. The errors measured (3 measures were used, called Fit, FPE, MSE) should be mapped back into uncertainties of the coefficients in the identified/fitted transfer function G to demonstrate that the delays are significantly different from zero (expressed via some confidence intervals). As the delays are so extremely small, they would likely be well approximated by a inverse linear expansion of the exponential function (exp(-s)=1/(1+s)), and could thus be incorporated into the denominator.

Author response: Thank you for your observations. We’ll address your three points in turn:

Point 1 (delay observability): We concatenated the tracking sections and the tracking data length in Fig. 5(c) of the revised paper is approximately 22 seconds. Each identification used recorded data exceeding 15 seconds.

A fixed time delay of time tau appearing in a control loop may be represented with a transfer function G(s) = e^(−tau*s), which has a Bode plot with unity gain and decreasing phase, as illustrated in the figure (right). At these low frequencies, the slope of this phase delay is nearly constant, which is helpful in the identification process. While we would love it if the insects had tracked the signals at higher frequencies, we were limited to the behaviors we could reliably induce, and the delay was nonetheless the most well-identified aspect, which is why it became a focus of the paper.

Point 2 (error/uncertainty): Parameter uncertainty analysis is a good idea, and we did find the uncertainty of the identified transfer function in terms of the identified parameter covariances–. For example, the covariance matrix of the identified parameters in the (1 zero, 2 poles, delay) case is showed in Table 1 (in Response to Reviewers file).

Table. 1: Covariance matrix of the identified zero, poles and delay parameters

(The diagonal term in a covariance matrix indicates the variance of each parameter.) We found a box plot of uncertainty variance from the entire data set, which is displayed in Fig. 1 

Fig 1. Variance of parameters uncertainty of model structures 

where the bottom and top represent the 25th and 75th percentiles, respectively, and the center point is the median. Although we computed and studied these, we do not consider the uncertainties to be of particular interest because the focus of this study is to determine how time delays in the agents affects swarm behaviors. Please see also the response of R1C2. 

Point 3 (delay model structure): We considered the delay to be pure tracking delay. The delay term does not change the gain in the transfer function but adds phase proportional to frequency. We appreciate your suggestion to also consider approximating the delay terms by using the linear expansion. We have now added this method to the identification and compared the fit results over all trajectories. 

The table (in Response to Reviewers file) shows that most (36) of the trajectories considering pure delay term give the better FIT percentage. The average of FIT percentage is 81.45 %. On the other hand 14 trajectories give better FIT for linear approximation. Having verified this, we are confident that the pure delay model gives the best result and the paper continues to rely on it; however, we have expanded the method to include this additional analysis.

Author action: In the revised version, we have added this comparison in Table. 4, and expanded the system identification section to consider both delay models.

R1C2: The connection between the experiments performed to gather data and the simulations is tenuous. The identified transfer function for each bee implies a certain (linear) dynamical system (2nd or, mostly, 3rd order, with delay). But it appears that this linear model was not used in the model at all except for the delay. It would also be a stretch to argue whether the delays and transfer function coefficients measured for an isolated bee (is that correct?) tracking a light strip can be applied to a bee's dynamics in a swarm, where it would have to track the various targets represented by the repulsive and attractive potentials (eqs 24, 25). 

A question related to the tenuous connection: why are 2d models used when the experiment goes to great length to extract a 3d trajectory?

Author response: 

Point 1 (identified transfer function): Thank you for the important points you mentioned. Yes, the delays were only used on the swarm model. In our experiment the identified dynamic system is not the primary outcome of the study. One reason for that is that the dynamics may show sensitivity to the tracking threshold; eg, if one considers only sections with perfect tracking, the transfer function in this ideal case would approach gain close to 1. In that case, only the phase delay associated with the visual reaction time remains. To provide robustness to the tracking threshold, the identified delay is the focus of this study. Numerous previous studies focusing on insect tracking data also focus on identifying tracking delays exclusively ([https://royalsocietypublishing.org/doi/10.1098/rstb.2016.0078, https://citeseerx.ist.psu.edu/viewdoc/download?doi=10.1.1.882.6246). There is a significant amount of analysis remaining to fully describe the delayed behavior. 

However, your suggestion to understand whether the identified dynamics support the same conclusions is helpful, and we followed it to incorporate the identified transfer functions in the simulation. We passed each agent’s integrated position in the identified dynamic system G_e(s). The overall behavior of the swarm remains, with some changes in the swarm center position. Our basic idea was that when multiple bees fly together they have heterogeneous delays (meaning of delays variation). It is difficult to say that the bees follow this swarm model having attractive and repulsive potentials, in fact we have related work showing the astonishing degree to which previously published swarm models fail to predict the recorded trajectories of insect swarms. At this early stage of swarm system identification, optimizing these poorly understood model structures will require a focused effort that has removed the effects of delays. Accordingly, we concentrated on how the measured delays could influence an example of swarm behavior.

Point 2: The swarm dynamic model may be readily posed in both 2d and 3d cases with no meaningful difference. As we solved delay differential equations using Matlab dde23 functions for many agents, the simulation time to complete the scope of this study in 3D exceeds workstation capabilities and would have required engaging high performance computing, without any expected theoretical difference. Instead we showed the 2d simulations in the manuscript for easier computation. Nonetheless, to verify this is the case, we have now completed several of these 3D simulations, and the behavior remains as expected.

Action taken: We have incorporated the identified dynamic systems with the delays in a new simulation section. We added a subsection “Effect of delays and transfer function” in the simulation section to show the results of adding transfer functions in the simulation. 

In the revised manuscript, we clarified that the dynamic model also applies for 3D case in the analysis and simulation section, and we added Fig.11 in the appendix section to illustrate the 3D behavior.

R1C3: The description of the data processing steps is exceedingly difficult to follow. There is a large number of inconsistencies in notation and there is an unevenness in the level of detail given (some steps are described in great detail while other, equally important ones, are jumped over). In my list of minor issues I list a few of the points where I had difficulty following. The list is not complete, though, such that the authors should revise this section aiming for more clarity.

Author response: Thank you for your efforts to understand the data processing steps despite having ambiguities. 

Action taken: We have rewritten the data processing steps to make it easier and balance the level of detail throughout the whole section. We have edited most of the notations and made a symbol list in the updated manuscript. We have also moved the “Kinematic tracking” section in the appendix.

Minor issues:

R1C4: p5,eq1: is T_j(m) in R^3 or is this a number? Further, Delta T appears to be a function of a single integer below eq1, but of g+1 integers in eq1. THis makes the definition of S_{g x 2} unclear.

Author response: Thank you for catching this issue. Here T_j(m) is a number not R^3. T_j (m) is the mth point of the X coordinates of stimulus and bee. Yes, Delta T is a function of a single integer. However, in equation 1 the output of Delta T is a vector of g+1 integers for both the stimulus and bee. If there are g points in a section, the matrix will have g rows and 2 columns. 

Action taken: We have rewritten the equation to avoid confusion.

R1C5: In eq2 D(i,j) appears on both sides of the equations, so the equation should be simplified to D(i,j)=E(i,j). I suspect, however, that the D(i,j) on the left-hand side is the D(i,j) of the next time

step. It would be helpful to include time as an argument because that would also clarify if in eq3 the updated D(i,j) is used or the D(i,j) from the previous time step.

After the explicit formulas (2-4) a single sentence "apply Gaussian blur and morphological closure" skips over much more substantial transformations. 

Author response: Thank you for your suggestions to improve equation 2. 

Yes, the updated background D(i,j) and current frame E(i,j) are used in equation 3. 

Thanks for noticing the later part where details of the Gaussian blur and morphological closure are absent. 

Action taken: We have added the time argument in equation 2 and also clarified the equation 3. We have also written some sentences regarding Gaussian blur and morphological enclosure and details how we applied these techniques. 

R1C6: Eq7 has several strange features. It appears to be a differential equation, dot x(t)=Ax(t-1)+Bu(t), with an enormous delay equal to 1 (measured in seconds?). The quantities p_t and v_t are not explained and the relationship between x and p,v is not defined. I suspect that eq7 is not a differential equation but a time step of length delta t. Correspondingly, should the first term be A x(t-delta t)? A similar correction may have to be applied to eqs (5,6)?

Author response: Thank you for your effort to understand the equation 7. This is a discrete Kalman filter model. Here the x(t-1) is the state at one step previous time. Yes, the relationship between x and p, v was absent. The correction in equation 5 and 6 should also be applied. 

Action taken: Kinematic tracking section was rewritten to avoid any confusions. We have added the discrete time step argument k instead of time t in the equation 5, 6 and 7. The relationship between x and p, v was showed. Equation 5, 6 and 7 are also rewritten also. 

R1C7: The term u in eq7 is never defined. I suspect that the Kalman filter is applied to the model of straight flight where measurement data corrects the model (in effect providing the acceleration a(t))? I could not understand the description. If eq7 is not a differential equation, one may have to use the Kalman filter for discrete time stepping. 

Author response: Thank you for your observation. Yes, the u was not defined explicitly. Your assumption is right. This is basically acceleration. Equation 7 is not a differential equation, it is the state space form of the states. We realize it should be clearer.

Action taken: In the revision, the Kinematic tracking section is rewritten considering the above mentioned issues. We rewrote equation 7 in a form that we believe will be easier to understand. 

R1C8: In eq15 the notation S_3^j is unclear, S_j appears to be a row of a 3x4 matrix, so is a 1x4 (row) vector. How can one take a power of this vector? The term P_2^j is not defined at all. 

Author response: Thank you for your patience for trying to understand. S_3^j is the 3rd row of calibration matrix S of camera j. Yes, S_3 appears to be a row of 34 matrix and it is 14 (row) vector. I am afraid if you find power in this (S_3^j) notation. Here, j simply indicates the camera number. As it has multiple cameras, so the 3rd row of each calibration matrix is different. The term P_2^j is a typo. It should be S_2^j which means 2nd row of j camera’s calibration matrix. 

Action taken: We have fixed the typo and written a clearer description of the notation S_3^j, S_2^j to avoid any misinterpretation. 

R1C9: Fig5c seems to suggest that the windows in which the bee is determined to be tracking are glued together before further data processing. Is this justified? This could have a large effect on the delay estimate. 

Author response: Thank you for this point. The bees track the stimulus intermittently. To increase the amount of data contained in each analysis, we did concatenate trajectories. This is a standard technique in system identification (described in textbooks by Tischler & Remple, Klein & Morelli, and Ljung, for example), which when done appropriately, will provide more data points and improve the identification. The justification is more straightforward in the frequency domain, while time domain approaches have more sensitivity to discontinuities. In this study, the stimulus and bee points were found from the images at the same time, reducing the impact of stimulus and bee’s trajectories discontinuities. 

R1C10: The equations on p8 contain several undefined terms: G(s) is the transfer function of what? Later G(s)e^(st au) is mentioned without specifying what tau is.

The minimisation in eq18 is not clear: what class of G_e is minimised over and what is tau? 

Author response: Thank you for catching this. Here G(s) is the transfer function of the stimulus and bee trajectory obtained from their frequency domain transformation. The G(s) and time delay tau are not known. Our goal is to estimate the transfer function and time delay. 

Our system identification approach finds the best possible transfer function G_e and time delay tau_i by minimizing the difference between measured and estimated frequency domain data.

Action taken: In the revised manuscript, we have tried to remove the confusion by editing the equation 18 and rewritten the section. 

R1C11: which norm is used in eq(19)? 

Author response: Thank you for the question. This is L-2 norm or Euclidean norm. 

Action taken: We have added the norm type in the description in the revised version. 

R1C12: there is no index i in the terms of the sum in eq20. 

Author response: Thank you for the observation. Yes, we should add the index i in the equation. 

Action taken: We have rewritten the equation by adding the index i, now in equation 10.

R1C13: Is the quantity e a function of two variables (t,theta) as suggested in eq(21) or one variable (t) as seen below eq(21)?

Author response: Thank you for the question. Here e is a two variable (t, theta) function. 

Action taken: In the revised version, we have written e as two variable function everywhere. And we extended the description of the formula to make it easier to read. 

R1C14: Below eq(27) X_i is is a vector, as established in the sentence. But then X_i shows up in the denominator defining cos theta. How do we divide by a vector here?

Author response: Thank you for pointing this issue. We forgot to give norm sign there. It should be L2 norm of X_i. So the denominator of the cos(theta) and sin(theta) would be (1/N)*||X_i||. 

Action taken: We have solved this issue by putting norm sign in the denominator of cos and sin variable in equation 27 (now equation 18). 

R1C15: In eq(26) a quantity epsilon shows up, while in eq(24) we have rho. Are they the same? 

Author response: Thank you for identifying the typo. Yes, both are same. So in the equation 26, it should be rho.

Action taken: In the revised manuscript, we changed epsilon to rho in equation 26 (now equation 17). 

R1C16: In eq(29) sup_{t>0} V_m(t)=0 is the same as demanding that V_m(t)=0 for all t>0 (since V_m is non-negative). Do you mean lim sup_{t\\to\\infty} V_m=0? 

Author response: Thank you for your observation. Yes, your assumption is right. As time goes to infinity the V_m tends to zero. It makes more sense if we put a limit there. Then we believe the supremum term does not require.

Action taken: We have added limit in the velocity convergence term of equation 29, now in equation 20. 

R1C17: In eqs(31,32) the undefined quantities V and X show up. Are all these quantities (U,V,X) implicitly having the index i,j? 

Author response: Thanks for the observing the undefined terms. We apologize for the typos here. All X should be U and it is the distance between agent i to agent j. Yes, U and V have implicit i,j and i index respectively. 

Action taken: In the revised manuscript we have corrected the typos. In the equation 32 (now equation 42), we replaced X by U term. As V is the velocity, we replaced it with v_i. 

R1C18: It is unclear how the delays are taken care of in the stability computation in eq(36). They appear to have taken into account in the final results (as they make a difference in stability). Again, as the delays are small, the approximation x(t-tau) ~ x(t) - tau x'(t) would likely be accurate compared to the experimental uncertainty. 

Author response: Thank you for mentioning an important point. Here the linear stability of local behavior using the bifurcation parameter indicates stable and unstable equilibria. This theoretical analysis is not sufficient to fully characterize how delays affect the behavior. We only can get different eigenvalues (stable and unstable) of the Jacobian matrix which produces swarm patterns all together. Instead, we applied the experimental delays in simulation to determine how delays affect swarming behaviors.

Action taken: We have added few sentences in the linear stability section mentioning the idea to use bifurcation and how delays can affect global behavior. 

R1C19: For the simulation the delays are assumed to have Gaussian distribution. This does not make sense as it would imply that delays could be negative. Sigma not much smaller than mu. 

Author response: Thank you for your point. You can see in the Fig 7c, the Gaussian distribution fit (mean 22 ms, standard deviation 40 ms) shows positive delays part on the experimental histogram. For our simulation we have only taken the positive delays from Gaussian distribution.

Action taken: We edited the definition of Gaussian distribution N(tau|mu,sigma). In the definition we have clearly written about the positive delays taken in the simulation and changed the notation N(tau>0 |mu, sigma) to make it correct. 

Reviewer 2

R2C1: In the section “Kinematic tracking”, the control input u(t) is not defined. It seems to be the acceleration, but this is not explained. 

Author response: You are correct, the u was not defined explicitly, and it is acceleration. As detailed in the response to R1C7, we have corrected this.

R2C2: Equation (7) is a discrete-time system with time-step $\\delta t$, yes? If so, the time-derivative over-dot is probably a typo? Also, only two spatial dimensions are assumed, why not three? The latter fact is explained later, but commentary should be added in this section to orient the reader.

Author response: I appreciate your observation.. Yes, equation 7 is a discrete time system. This is a typo: time derivative. The Kalman filter algorithm is applied the 2D centroid points in each camera’s image plane coordinates, hence the 2-D points.

Action taken: We have rewritten the kinematic tracking section describing the discrete Kalman filter. Typos have been fixed in the revised version. In response to R1C3, the kinematic tracking section has also been moved to the appendix.

R2C3: Since you describe noise, Equations (6) and (8) should presumably have additive Gaussian white noise terms for example, per the canonical formulation of Kalman filtering. 

Author response: Thank you for suggestions. We considered Gaussian noise in the process and measurement noise covariances which have effects on the prediction and correction steps of Kalman filtering. 

Action taken: We have reworked the section on Kalman filtering in the updated manuscript. 

R2C4: Where does the state-space noise come from in Equation (9)? There needs to be a noise amplitude somewhere, like in E_{z}. 

Author response: Thank you for the question. The process noises E_x and E_z come from the standard deviation of position and velocity which can be obtained from the standard deviation of acceleration σ_a multiplied by (δt)^2/2 and (δt) respectively. 

Yes, there should be noise amplitude with the time step in the process noise in equation 9. 

Action taken: We have updated the description of how we applied these process noises in the Kalman filtering portion of the revised manuscript. We have added the standard deviation of acceleration sigma_a in equation 9 (now in equation 33). 

R2C5: The notation in Equation (11) is confusing. Shouldn't the a priori estimate at the current time depend on x^{+}(t-\\delta t): the aposteriori estimate at the past time...? Overall the mix of discrete and continuous in this section is unclear.

Author response: Thank you for your effort to understand the equation despite having continuous and discrete mixing. Yes, the apriori estimate at the current time x^-_k should depend on x^+_(k-1) the a posteriori of the past time and the acceleration term. 

Action taken: We have rewritten the equation using only discrete term in the revised version. It is now presented in equation 36. 

R2C6: There is a parentheses missing in Equation (13). 

Author response: Thank you for catching this point. 

Action taken: We added the missing parentheses.

R2C7: In Equation (16), the notation is confusing. Is "i" imaginary or "j"? 

Author response: Thank you for noticing this. 

Action taken: Both i and j are imaginary in equation 16. To avoid confusion we have written consistent j for imaginary in both places. 

R2C8: Just above Equation (17), is the assumption of linearity (between visual stimulus and trajectory-response) a good approximation? If so, how was that quantified?

Author response: Thank you for your response. Coherence measures the degree of linear dependency of two signals by testing for similar frequency components. If two signals correspond to each other perfectly at a given frequency, the magnitude of coherence is 1. If they are totally unrelated, coherence will be 0. For this analysis, a coherence value exceeding 0.7 (Remple/Tischler textbook) was used to identify stimulus and bee trajectories having a linear relationship. 

Action taken: In the revised version, we have added details of coherence before this equation to make clearer to the reader. 

R2C9: Equation (21): This is a general formula. It would be more interpretable, if you used the actual variables in the current fitting/estimation problem. 

Author response: Thank you for your observation. We have considered an ARX (autoregressive with exogenous input) model as x[n] + a_1*x[n-1] +...a_n*x[n-n_a] = b_1*u[n-1] + b_2*u[n-2]+....b_n*u[n-n_b] + e[n], here e[n] is the white noise term.

The estimated parameters theta = [a_1 a_2 …..a_n b_1 b_2 ……b_n]^T. The regression matrix is phi[n] = [-x[n-1] …. -x[n-n_a] u[n-1] …. u[n-n_b]]^T. The prediction error can be written as e(t,\\theta_J) = x[n] - \\phi[n]^T*\\theta. Then the FPE can be obtained from the equation 21. 

Action taken: In the revised manuscript we have added all these details of the equation 21 (now equation 12) which we think is more interpretable. 

R2C10: There is no self-propulsion in Equations (22-23). Hence, if no neighbors are present, then the agents *stop* due to friction. Is that right? Without some tendency to keep moving, the “swarming” system seems like an inert, equilibrium system, not an active matter system relevant for swarming…Please comment. 

Author response: Thank you for your comment. Yes, there is no self propulsion term in the swarm model. We considered only the friction term and interaction among each agent to move forward. Hence, if there are no neighboring agents, agents will not move. 

R2C11: What is x_{ij}....? It is defined implicitly, though ambiguously, in Equation (25). 

Author response: Thank you for getting this point. The term x_{ij} is denoted by x_i(t) - xj(t-\\tau_{ij}).

Action taken: We have written explicitly the term x_{ij} to avoid any ambiguities. 

R2C12: In Equation (24), shouldn't each quantity within the "sums" be squared, individually? Ditto with the fourth power. Typically, potential functions (at least in physically-inspired models) are pairwise. This phrasing seems to generate cross terms, which is a bit strange. 

Author response: Thank you for the observations. Yes, typically aggregation potential function use interagent distance as a function of exponential or power law (different powers). However, in our case as we have calculated at first the sum of inter agent distance then subtracted r from that. This whole term is then second and fourth powered to pose the attraction. We see the cross terms do not provide any problems because equations 17 and 18 show how the potential function uses the positions of the agents.

R2C13: In the section "Long range attraction,” the derivation seems like an unnecessary use of formalism. The repulsive potential is exponentially damped, whereas attraction (for example) is infinite range-- the polynomial functions grow to infinity as the distance does. I would just state the fact that you are looking in the regime of long-range attraction where pairwise distances are >> Dr. 

Author response: Thank you for the observations. Your assumption is right that the repulsive potential is exponential damped which means when the distance is large it would tends to zero. We could only state this fact however, we thought it would be helpful to prove this using some mathematical argument. 

Author action: We have moved this section to the Appendix.

R2C14: X_{i} being constant means that agent "i" is a constant distance from the swarm center of mass, yes? If so, add some commentary and explanation, etc. near Equation (36). 

Author response: Thank you for the observations. Here, agent “i” is in a constant distance means it has constant distance from the origin. 

Author action: We have added some comments for this near equation 36 (now equation 24). 

R2C15: The Jacobain for the full swarm should have N*times the dimension of Equation (36). It seems like you are studying the linear stability with respect to perturbations in each agent (treated separately, as localized perturbations)...Probably, this is the same as a mean-field analysis, which is sometimes accurate and sometimes not. To judge, we need to see comparisons to full-swarm simulations. On that point, see for instance https://link.aps.org/doi/10.1103/PhysRevE.101.042202, which formally analyzes delay effects in self-propelled swarming systems. The authors should consider citing this work since it addresses delay-induced instability in swarms beyond the mean-field analysis that they present, and would give the present work some cover in terms of pointing to more detailed treatment, etc. 

Author response: Thank you for your valuable insights. Yes, full swarm would contain the agent number N* times the equation. However, we only analysis here the individual level behaviors which does not include N. Using mean field analysis we may not predict the swarm behaviors as a whole accurately when time delayed effect arise. This paper is really a helpful to see the delay effects on ring and rotational patterns of a swarm model. 

Action taken: We have cited this paper in the revised manuscript. 

R2C16: Where does time delay enter into the picture (in the Linear Stability section)? It is not clear at this point. In fact, time delay will not just change stability of the solutions mentioned, but may change the solutions all together. If this is neglected, fine, but the assumption needs to be stated.

Author response: Thank you for your response. Yes, it is neglected here. Linear stability analysis only shows that bifurcation parameter can generate three swarm shapes-ring, cluster and double ring by developing stable and unstable modes. The time delay effect on the swarm is not considered in the linear stability analysis. The time delays play a crucial role to make the swarm stable or unstable all together which is showed in the simulation. 

Action taken: In the revision we have stated this assumption clearly in the linear stability section. 

R2C17: The section “Results and Discussion” has almost no commentary. It is disorienting to read. The figures are just plopped down, one after another. Please consider rewriting this section. 

Author response: Thank you for the suggestions. We wanted to show the step by step procedures. 

Action taken: In the revised manuscript, we have edited the section. 

R2C18: It seems like time-delay effects were only simulated in the swarming model, not formally analyzed (e.g., in the linear stability calculation). Is that correct? For one, if delay was included in the stability analysis, we should see an infinite dimensional stability spectrum, satisfying a transcendental equation with an infinite number of solutions. 

Author response: Thank you for your observation. Yes, your assumption is right. The time delay effect on the swarm was considered in simulation, not analyzed theoretically. Yes, one possible way to consider delay in the stability analysis of a swarm model is via the characteristic solution of the linearized system. In this case, this approach will yield a transcendental equation that contains an infinite number of solutions.

---

## [Decision Letter · Decision Letter 1]

11 Nov 2022

Experimental identification of individual insect visual tracking delays in free flight and their effects on visual swarm patterns

PONE-D-22-10206R1

Dear Dr. Islam,

We’re pleased to inform you that your manuscript has been judged scientifically suitable for publication and will be formally accepted for publication once it meets all outstanding technical requirements.

Kind regards,

Antony R Humphries, Ph.D.

Academic Editor

PLOS ONE

Additional Editor Comments (optional):

Reviewers' comments:

Reviewer's Responses to Questions

**Comments to the Author**

1. If the authors have adequately addressed your comments raised in a previous round of review and you feel that this manuscript is now acceptable for publication, you may indicate that here to bypass the “Comments to the Author” section, enter your conflict of interest statement in the “Confidential to Editor” section, and submit your "Accept" recommendation.

Reviewer #1: All comments have been addressed

Reviewer #2: All comments have been addressed

2. Is the manuscript technically sound, and do the data support the conclusions?

Reviewer #1: Yes

Reviewer #2: Yes

3. Has the statistical analysis been performed appropriately and rigorously? 

Reviewer #1: Yes

Reviewer #2: Yes

4. Have the authors made all data underlying the findings in their manuscript fully available?

Reviewer #1: Yes

Reviewer #2: Yes

5. Is the manuscript presented in an intelligible fashion and written in standard English?

Reviewer #1: Yes

Reviewer #2: Yes

6. Review Comments to the Author

Reviewer #1: The authors have convincingly responded to the major issues I raised in their "Response to Reviewers" and incorporated corresponding imporvements into their manuscript. All concrete minor issues raised have been addressed.

Reviewer #2: The authors have addressed the comments and critiques of both referees. The manuscript is significantly more clear and easy to follow. I still think that two distinct papers, in this case, for theory and experiment might be better, but I understand the authors' arguments for combining them. Hence, I recommend publication.

7. PLOS authors have the option to publish the peer review history of their article (what does this mean?). If published, this will include your full peer review and any attached files.

Reviewer #1: No

Reviewer #2: No

---

## [Editor Report · Acceptance letter]

16 Nov 2022

PONE-D-22-10206R1 

Experimental identification of individual insect visual tracking delays in free flight and their effects on visual swarm patterns 

Dear Dr. Islam:

I'm pleased to inform you that your manuscript has been deemed suitable for publication in PLOS ONE. Congratulations! Your manuscript is now with our production department. 

Kind regards, 

on behalf of

Prof. Antony R Humphries 

Academic Editor

PLOS ONE